# Sorghum (*Sorghum bicolor* L. Moench) Gluten-Free Bread: The Effect of Milling Conditions on the Technological Properties and In Vitro Bioaccessibility of Polyphenols and Minerals

**DOI:** 10.3390/foods12163030

**Published:** 2023-08-12

**Authors:** María Isabel Curti, Pablo Martín Palavecino, Marianela Savio, María Verónica Baroni, Pablo Daniel Ribotta

**Affiliations:** 1Facultad Ciencias Exactas y Naturales, Universidad Nacional de La Pampa, Santa Rosa 6300, Argentina; mariacurti@gmail.com (M.I.C.); msavio@exactas.unlpam.edu.ar (M.S.); 2Instituto de Ciencias de la Tierra y Ambientales de La Pampa (INCITAP, CONICET-UNLPAM), Santa Rosa 6300, Argentina; 3Instituto de Ciencia y Tecnología de Alimentos Córdoba (ICYTAC, CONICET-UNC), Córdoba 5000, Argentina; pmpalavecino@gmail.com (P.M.P.); vbaroni@unc.edu.ar (M.V.B.); 4Facultad de Ciencias Exactas, Físicas y Naturales, Universidad Nacional de Córdoba, Córdoba 5000, Argentina; 5Facultad de Ciencias Químicas, Departamento de Química Orgánica, Universidad Nacional de Córdoba, Córdoba 5000, Argentina

**Keywords:** sorghum, milling, flour, gluten-free, bread, technological properties, nutritional characteristics

## Abstract

The absence of gluten proteins in sorghum allows for the production of baked goods that are suitable for celiacs. Previous studies have shown that the milling process affects the performance of sorghum flour in baked products, especially those that are gluten-free (GF). This study aimed to explore the effects of mill type (impact and roller) on flour properties and GF bread quality by assessing the technological quality, antioxidant activity, and mineral content of the bread. All particle populations of flour obtained via both millings presented a bimodal distribution, and the volume mean diameter (D 4,3) ranged from 431.6 µm to 561.6 µm. The partially refined milled flour obtained via polishing and impact milling produced bread with a soft crumb, fewer but larger alveoli in the crumb, and a structure that did not collapse during baking, showing the best performance in bread quality. In the in vitro bread digestibility assay, the total polyphenol content and antioxidant activity decreased during the digestion steps. High mineral (Cu, Fe, Mn, and Zn) contents were also found in a portion of the bread (120 g) made with whole sorghum flour; however, their potential bioavailability was reduced in the presence of a higher amount of bran.

## 1. Introduction

The elaboration of gluten-free (GF) bread is challenging due to the need to replace the functionality of the gluten network. The utilization of GF flour (e.g., rice, maize, sorghum, buckwheat, amaranth, quinoa, corn) in breadmaking inhibits the development of a viscoelastic dough, which is essential for retaining the carbon dioxide generated during fermentation and baking. Another limitation associated with GF flour and its performance in breadmaking is the rapid onset of staling. Most gluten-free doughs tend to contain higher water levels and show a more fluid structure (batter), making bread production more complex [1]. The plasticizing effect of water is critical in the hydration and gelatinization of starch granules during baking, which is important for the development of bread with a desirable structure and texture [2]. However, excess water can cause the excessive expansion of a loaf in the presence of large alveoli in the crumb [3]. The pasting properties of flour affect the rheological properties of batter and play a central role in determining the carbon dioxide retention capacity and texture of bread [4,5]. The starch in flour gelatinizes during the baking process and increases in viscosity, preventing the sedimentation of ungelatinized starch and the escape of air bubbles and gas [6]. In addition, the particle size and water retention capacity of flour must be considered, since large particles tend to have lower hydration capacities compared to fine particles [7,8].

The process of making gluten-free bread differs significantly from the standard process used for wheat bread. Traditionally, wheat dough is mixed, divided, and shaped; it is then left to rise and finally baked. A method of producing gluten-free bread with good characteristics consists of mixing, proofing, and baking [9]. This method has been successfully applied in subsequent studies on gluten-free bread [10,11,12,13]. A lack of structure, such as the structure generated by gluten, and the presence of repulsive forces between starch granules are responsible for the lack of stability in bread systems [13]. As a result, gluten-free bread is stiffer, and its texture is more irregular and crumblier due to irregular and unstable alveoli produced by a lower gas retention capacity [14]. To overcome the lack of a gluten network, gluten-free bread formulations involve various approaches to improving their technological, sensory, and shelf-life properties, such as the combination of different cereal flours (rice, corn, and sorghum) [13,15,16], pseudocereal flours (quinoa, amaranth, and buckwheat), and leguminous flour [17,18]; cereal and tuber starches [19,20]; hydrocolloids [21]; and emulsifiers and industrial fat [22]; or their combination [23].

The formulation of gluten-free products also continues to be a challenge from a nutritional perspective. Since wheat-based food is an important source of energy intake, it has been shown that people, particularly children, who follow a strict gluten-free diet often suffer from nutritional deficiencies [24]. Malabsorption is a hallmark of celiac disease, leading to deficiencies of calcium and other minerals. In addition, several commercially available gluten-free products show lower levels of folate, iron, and B vitamins or are not enriched or fortified when compared to wheat-based products [25,26].

Although the inflammation produced in the mucosa is controlled via the total elimination of gluten, the incorporation of anti-inflammatory agents is believed to offer benefits not only at the local intestinal level but also at a systemic level. In this sense, sorghum is rich in phytochemicals, ps [27]. All sorghum varieties contain phenolic compounds, although the types and levels present are related to the color of the pericarp and the presence of a pigmented testa [28]. Most sorghum tannins are condensed and consist of oligomers or polymers of catechins (flavan-3-ols and/or flavan-3,4-diols). Regarding flavonoids, three classes are found in large proportions: anthocyanins, flavones, and flavonones [27]. These compounds may protect against oxidative stress, UV exposure, and pollutants that contribute to the development of chronic diseases such as type 2 diabetes, heart disease, and some cancers [29].

The use of sorghum in the preparation of gluten-free bread has been analyzed in combination with other types of flour. Schober et al. [13] used decorticated white and red sorghum flour in a 70/30 ratio (sorghum flour/corn starch) and reported varied results depending on the amount of water and xanthan gum added. Velázquez et al. [30] made bread with sorghum and corn flour in variable proportions, developing products with acceptable technological characteristics with a 50/50 flour ratio. Trappey et al. [31] analyzed white sorghum flour with different extraction levels (70%) and unmodified potato starch (30%) to prepare gluten-free bread, observing better results at a higher degree of sorghum flour refinement. We found a few studies in which sorghum flour is the major component of gluten-free bread; hence, it is important to thoroughly study the effect of different types of sorghum flour on the quality of these products.

Sorghum flour is typically produced by grinding grains into a fine powder. The process usually involves several steps: cleaning, dehulling (partially or completely), milling, and packaging. Partially dehulling and milling have major effects on flour properties and product quality. Several types of mills can be used to produce sorghum flour. The most common types include stone mills, roller mills, hammer or impact mills, and disc mills. The selection of mill depends on several factors, including the desired flour properties, volume of production, availability of resources, and technology. It is known that the type of mill affects the performance of flour in baked products, especially those that are gluten-free [7,15,32]. In a previous study, the authors carried out a complete analysis of the influences of mill type and the grain tempering of different sorghum varieties on the physicochemical and functional properties of flour [33]; however, the effects on the technological and nutritional quality of bread remain to be studied. Improvement in the sorghum milling technology applied to food requires an in-depth understanding of how the type and milling process of grains affect flour properties and their performance in baked goods.

Thus, the goal of this study was to investigate the effects of mill type (impact and roller) on flour properties, the technological properties of GF breads, and the in vitro bioaccessibility of the polyphenols and minerals in GF breads. Samples of bread were made with three white sorghum flours: integral and polished (partially dehulled) flour, both obtained via impact milling, and flour obtained via roller milling with subsequent sifting. The flour fractions were characterized; the loaves of bread were developed and subsequently subjected to an in vitro digestibility test to evaluate their antioxidant properties and the potential bioavailability of the minerals they contained.

## 2. Materials and Methods

### 2.1. Flour Production

A commercial hybrid of white sorghum was provided by Praga S.R.L (Córdoba, Argentina). Sorghum flour samples were obtained according to the protocols described in previous work [32,33]. Briefly, the sorghum grains were cleaned (1500 g), conditioned at ~12% moisture, and allowed to rest for 24 h at room temperature before undergoing polishing and milling.

Roller milling: cleaned and conditioned sorghum samples were milled on a roller mill (Agromatic AG AQC 109, Laupen, Switzerland) and then sieved through a 30-mesh screen (595 μm) to separate the bran (WS-RM).

Pearling and milling processes: cleaned and conditioned grains were polished via abrasion, employing a laboratory rice dehuller (PAZ-DTA1, Limeira, Brazil). The polishing time was set at 180 s, according to previous experiments. The pearled samples were milled using an impact mill (Fritsch Pulverisette 16, Idar-Oberstein, Germany), outfitted with a 0.75 mm screen (PWS-IM).

An unpolished flour sample (wholegrain, containing the bran) was also obtained via impact milling (WWS-IM).

### 2.2. Flour Properties

Ash (dried basis) and moisture contents were determined via standard methods [34]. Analyses were performed in duplicate.

The particle size distribution (PSD) was assessed using a Horiba (LA 960, Irvine, CA, USA) laser light diffraction instrument with a dry dispersion module, employing an airflow of 0.40 MPa and a feeder speed of 75. The representative dry samples (12 to 15 g and approximately 6–8% moisture) were analyzed. The parameters obtained were as follows: D 4,3, D10, D50, and D90 (particle diameters for which the cumulative volumes of the particles were 10%, 50%, and 90%, respectively). The particle size polydispersity index or span index associated was determined as [(D90 − D10)/D50)]. Analyses were performed in duplicate.

The color of the sorghum flour was evaluated utilizing a colorimeter with a D65 illuminant at a 10° observer angle (CM-600d; Konica Minolta, Japan). The results were presented in terms of CIELAB parameters (*L*, a*, b**). The total color difference between flours (Δ*E*) [35] was calculated in relation to WWS-IM (WWS-IM to PWS-IM and WWS-IM to WS-RM), using the following equation:$$\Delta E=\sqrt{\Delta {L*}^{2}+\Delta {a*}^{2}+\Delta {b*}^{2}}$$

*L**, *a** and *b** are de CIELAB parameters.

The apparent viscosity profile of each sample was analyzed using a rapid visco-analyzer instrument (RVA series 4500, Perkin Elmer, Waltham, MA, USA), as described by Palavecino et al. [36]. The pasting parameters, including the pasting temperature (PT), peak viscosity (PV), final viscosity (FV), breakdown (BD), and setback (SB), were acquired using Thermocline for Windows© software (V 3.15, Perten Instruments, Ryde, Australia). The analyses were performed in duplicate.

### 2.3. Elaboration of Bread

Bread was made following the procedure outlined by Sciarini et al. [16]. The formulation was: 180 g of sorghum flour, 20 g of rice flour (Dimax, Córdoba, Argentina), 10 g of cassava flour (Dimax, Argentina), 6 g of commercial pressed yeast (Calsa, Buenos Aires, Argentina), 4 g of commercial margarine (Dánica, Buenos Aires, Argentina), 4 g of sodium chloride, 2 g of carboxymethyl cellulose (CMC) (Nicco distributor SRL, Córdoba, Argentina), 1 g of sodium stearoyl lactylate (SSL) (Sigma–Aldrich, Saint Louis, MO, USA), 0.2 g of chemical leavening agent (Nicco distributor SRL, Córdoba Argentina), and 200 mL of water. Briefly, the solid ingredients were mixed, and the yeast dispersed in the water was then incorporated. The ingredients were mixed in a planetary mixer (Peabody SmartChef, Argentina) for 1 min at a low speed and 1.5 min at a high speed. The batter was allowed to rest for 30 min. It was mixed again for 30 s at minimum speed. Portions (65 g) of the batter were poured into aluminum molds and subjected to a second fermentation for 60 min (30 °C and 85% relative humidity). Finally, they were baked in a convective oven for 30 min at 180 °C (Beta 107 IPA Oven—Pauna, Argentina). The batters and bread were produced in duplicate.

### 2.4. Characterization of Batters

The back extrusion test was performed with an INSTRON 3342 texture analyzer (USA), following Sciarini et al. [16] with some modifications. The batter samples were prepared as described for the baking process but without adding yeast and omitting the fermentation steps. A constant mass of sample (40 g) was poured into the extrusion vessel (diameter: 36.5 mm), and the air pockets were removed with a spoon. The batter sample was back-extruded via the cylindrical probe (diameter: 35 mm) at a speed of 1 mm/s and a distance of 25 mm. The extrusion force was recorded as a function of time. The extrusion force was obtained as the force required to extrude the sample back to 15 mm from its height. The batters were analyzed in triplicate.

### 2.5. Evaluation of Bread Quality

The bread volume was determined via seed displacement 24 h after baking. The specific volume (SV) was obtained by dividing the sample volume by its weight (cm^3^/g) (Method 10-05; AACC, 2010). Five determinations were made for each batch, and the average value of the two baking tests for each sample was reported.

The bread crumb color was determined using a colorimeter with a D65 illuminant at a 10° observer angle (CM-600d; Konica Minolta, Tokyo, Japan), with the results expressed using CIELAB parameters (*L*, a*, b**). Three slices of each bread were obtained, and five color measurements were made in each crumb, reporting the average value.

The total color differences (Δ*E*) between the WWS-IM crumb and the PWS-IM and WS-RM crumb were calculated as described for flours in Section 2.2.

The firmness (F) of the bread crumb was evaluated via an INSTRON 3342 universal texture analyzer (USA), as performed by Sciarini et al. [16]. Two 1.5 cm thick slices were obtained from the center of each loaf and subjected to a compression test (in two cycles) under the following conditions: compression cell, 5 kg; head speed, 100 mm/min; distance, 10 mm; load, 100 g; maximum deformation, 40%; diameter of the cylindrical compression probe, 25 mm. Crumb hardness was expressed as the force required to compress the sample to 25% of its original thickness. Cohesiveness (C) was also determined as the quotient between the work done in the second compression and the first compression, while elasticity was calculated as the quotient between distance 2/distance 1 (from zero to the maximum of each compression). Gumminess (G) (G:F × C), and chewiness (M) (M:G × E) were obtained via calculation. The breads were analyzed in triplicate.

Two slices (1.5 cm thick) were obtained from the central region of each loaf and scanned using a scanner (HP Scanjet G3010, Palo Alto, Ca, USA). The images obtained were evaluated via the Image-J 1.48a image analysis program (National Institutes of Health, USA). The color images were transformed into 8-bit images and examined in grayscale (0 black, 255 white). The selected images (total area: 4 cm × 4 cm) were segmented using a gray value to create binary images via the Iso-Data algorithm [37]. Pixels with a gray level below the threshold were shown in black and considered air (cell), and pixels with a gray level above the threshold were shown in white and considered objects (cell wall). The selected crumb cell characteristics comprised the total number of cells, total cell area, mean cell area, and the cell fraction (total cell area ratio). The grain uniformity was established by calculating the proportion of small to large cell counts (i.e., the ratio of the number of cells with areas lower than 4 mm^2^); higher values indicate a greater degree of crumb grain uniformity [38]. The breads were analyzed in triplicate.

### 2.6. Evaluation of the Content of Polyphenols and Their Antioxidant Capacity in Sorghum Flour and Bread

The total polyphenol content, ferric reducing activity, and radical-scavenging activity were assessed in extracts obtained from the two chosen flours and their corresponding breads, WWS-IM and PWS-IM. One hundred milligrams of flour or ground bread was extracted with 1 mL of a mixture of methanol (Sigma–Aldrich, Saint Louis, MO, USA)/water (70:30 *v*/*v*) with continuous agitation for 5 min at room temperature. Afterward, the extracts were subjected to centrifugation (Gelec, Buenos Aires, Argentina) for 10 min at 800× *g*, and the supernatants were gathered. This procedure was repeated twice more, and all supernatants were pooled, filtered, and stored at −80 °C until further analysis.

The total polyphenol contents (TPCs) of the bread and flour extracts were determined via the Folin–Ciocalteu method, using gallic acid (Sigma–Aldrich, Saint Louis, MO, USA) as a calibration standard [39]. Briefly, the samples were properly diluted and mixed with Folin–Ciocalteu reagent and a 20% aqueous solution of sodium carbonate. The absorbance was then read at 750 nm. The TPC was calculated using a calibration curve constructed with gallic acid (with a fit of R^2^ = 0.99). The results were expressed as milligrams of polyphenols (equivalent to gallic acid) per 100 g of dry weight sample. All samples were analyzed in triplicate. Blank samples were employed to discount the absorbance attributed to solvents and reagents.

The radical scavenging activity and the ferric reducing activity of the bread and flour extracts were determined using the ABTS (2,2′-azino-bis(3-ethylbenzothiazoline-6-sulfonic acid)) [40] and FRAP (ferric reducing antioxidant power) [41] methods, respectively. Briefly, 100 μL of the properly diluted extracts was mixed with the corresponding reagent, and the absorbance was measured at 593 nm and 734 nm, respectively. In both cases, the results were obtained from a calibration curve made using Trolox (Sigma–Aldrich, Saint Louis, MO, USA). In both methods, the R^2^ fit curve was greater than 0.98. The results were expressed in mg of Trolox equivalent per 100 g of dry weight sample. All samples were analyzed in triplicate. Reaction blanks (containing only reagents) were used for each type of sample (flour or bread extracts) to discount absorbance due to solvents and reagents.

### 2.7. Determination of Phenolic Profile

The phenolic profiles of the flour and bread were obtained via HPLC-DAD-ESI-MS/MS, as described by Blanco Canalis et al. [42], using an Agilent 1200 Series LC system (Agilent, Santa Clara, CA, USA) coupled with a DAD detector (Agilent 1200 Series, USA), along with an ESI source connected to a MicroQTOF II (Bruker Daltonics, San Jose, CA, USA) mass spectrometer (MS and MS/MS). The HPLC system was equipped with a binary gradient pump, a solvent degasser, and an autosampler (Agilent Series 1200 L, USA).

The identification of polyphenols relied on their retention times and exact, MS, and MS/MS spectra, and each identification was cross-referenced with authentic standards whenever accessible. In cases in which authentic standards were unavailable, a tentative identification was conducted using the exact mass and MS/MS, considering compound references in the literature [43,44,45,46].

### 2.8. In Vitro Digestion

The digestive process in the mouth, stomach (gastric), small intestine, and large intestine, including colonic fermentation, was simulated in four steps [47,48]. Initially, two grams of bread was mixed with 2 mL of salivary fluid (containing amylase), and chewing was emulated using an Ultra-Turrax T18 blender (Ika-Labortechnik, Staufen im Breisgau, Germany). The sample was then adjusted to a pH of 2.0 and incubated in a water bath with agitation for 2 h at 37 °C with 500 μL of porcine gastric mucosa pepsin to simulate gastric digestion. Afterward, we simulated digestion and absorption in the small intestine, using a solution of pancreatin and bile salts in 0.1 M NaHCO_3_ (pH = 7.5). This mixture was placed in a dialysis bag (with a 10 kDa cut-off). Next, the dialysis bag was immersed in 0.1 M NaHCO_3_ (pH = 7.5) and incubated in the dark with agitation at 40 osc/min for 3 h at 37 °C. The fraction that traversed the dialysis membrane represented the fraction accessible for absorption into the circulatory system via passive diffusion in the small intestine. The remaining solution inside the bag was designated the non-dialyzable fraction. The dialyzed samples were filtered, fractionated, and stored at −80 °C until analysis. The non-dialyzable fraction was subjected to simulated colonic fermentation by adding 200 µL of an inoculum prepared with mouse fecal matter and placed back into the dialysis membrane, which was immersed in NaHCO_3_ (pH = 7.5). It was left under agitation at 40 °C in the dark for 24 h under anaerobic conditions. Finally, the dialyzable fraction (which passes through the dialysis membrane) was recovered and stored at −80 °C until analysis. The in vitro digestion procedure was conducted in duplicate. The total polyphenol content (TPC) and antioxidant properties (FRAP and ABTS) were determined in samples obtained from the in vitro assay after the intestine digestion and colonic fermentation steps, as described previously.

### 2.9. Mineral Determination

Cu, Fe, Mn, and Zn contents were determined in the bread samples before the in vitro digestion test (extract) and in the fraction corresponding to the potential absorption in the small intestine. A microwave-induced plasma optical emission spectrometry (MIP OES) Agilent MP 4210 instrument with axially viewed configuration (Santa Clara, CA, USA) was used for the analyte determination. The MIP OES includes a Czerny–Turner monochromator with a VistaChip charge-coupled device (CCD) array detector, operating online with a nitrogen generator. An SPS3 auto-sampler system, a One Neb inert nebulizer, a single-pass cyclonic spray chamber, and a quartz plasma torch (Agilent, Mulgrave, Australia) were used. The emission line wavelengths selected for the analytes were Cu (24.754 nm), Fe (371.993 nm), Mn (403.076 nm), and Zn (481.053 nm).

Extracts from the bread samples that were not subjected to in vitro testing were obtained via microwave-assisted digestion. Here, 0.50 g of dry sample was weighed and transferred to a hermetically sealed 10 mL PTFE tube. Then, 5.0 mL of HNO_3_ (Sigma–Aldrich, Saint Louis, MO, USA), 7 mol L^−1^, and 1.00 mL of H_2_O_2_ (Sigma–Aldrich, Saint Louis, MO, USA) were added. The tubes were placed in a microwave system (Anton Par, Austria), and the heating program was implemented as follows: 15 min for ramping up to 190 °C and up to 1200 W, and 15 min for maintenance time at 190 °C up to 1200 W. Then, the samples were placed in volumetric tubes and diluted up to a volume of 30.00 mL. The digestions were performed in duplicate. 

The samples obtained from the in vitro digestion process were filtered with filter paper (Whatman grade 1) to eliminate solids in the suspensions and later evaluated in the same way as the extracts obtained using the microwave system. Dialyzability (%) was used to estimate the compounds that would be available for absorption in the small intestine. It was calculated using the following equation: Dz (%):Y×100/Z. Here, *Y* represents the mineral element content outside the dialysis tubing (mg of mineral element/100 g of bread), and *Z* is the mineral element content in the bread sample (mg of mineral element/100 g of bread).

### 2.10. Statistical Analysis

The data were statistically treated using an analysis of variance (ANOVA). The means were compared via the LSD Fisher test at a significance level of 0.05 (*p* ≤ 0.05). For these analyses, InfoStat statistical software (Version 13p) was used (Universidad Nacional de Córdoba, Córdoba, Argentina).

## 3. Result and Discussion

### 3.1. Sorghum Flour Properties

The WWS-IM and WS-RM samples had the highest and lowest milling yields, respectively (Table 1). These yields were similar to those obtained in previous studies [32,33]. The viscosity profiles of the flours increased significantly as the milling yield decreased. The WS-RM sample showed the highest PV and FV and the lowest TP (*p* < 0.05) (Table 1). In agreement with our previous study [33], the peak viscosity (PV) and final viscosity (FV) increased with the refinement of the flour, which was attributed to a reduction in the outer layers (a lower ash content, Table 1) and germ, thus increasing the amount of starch available for gelatinization. The viscosity of a paste increases via the swelling of granules and the leaching of amylose due to heating and mechanical stress. Then, the formation of gel occurs via molecular rearrangement, increasing the number of binding sites. According to different authors [32,36,49], a high batter viscosity is desirable, especially in gluten-free baked goods like cakes and bread, since it results in enhanced air bubbles and CO_2_ retention.

All samples showed a bimodal particle distribution (Figure 1). They evidenced a large and broad peak with a maximum diameter of around 600 μm, suggesting the presence of particles including endosperm pieces and bran fractions (Appendix A). The samples also showed a minor peak with a diameter of approximately 25 μm; this was associated with the starch granule fraction., which normally varies between 2 and 30 μm [50]. The PWS-IM exhibited the highest D4,3; however, the differences were only significant concerning the WS-RM. Al-Rabadi [51] reported a D4,3 value similar to those obtained in this study (525 ± 20 μm) for sorghum flour produced via impact milling.

The polydispersity index value (span) decreased with the degree of refinement of the flour (Appendix A). The WWS-IM sample exhibited the highest span value, indicating greater heterogeneity in the particle size of the sample, which was linked to the presence of bran (Table 1, ash content). According to Zhao and Ambrose [52], the bran fraction of the sorghum grain has a more flexible and resistant texture (in contrast to the endosperm) which endures with increased humidity, making the grinding process difficult.

Regarding the color of the flour (Table 1), *L** increased after bran removal. The PWS-IM sample obtained a higher *L** value than the WS-RM sample, which is related to its lower ash content. Similarly, the redness (*a**) decreased as the ash content decreased, with the WS-RM sample obtaining the lowest value. Yellowness (*b**) did not show significant differences between samples.

The Δ*E* values represent the Euclidean distance between one color and another, and a scale correlating the Δ*E* value and visual perception capacity was established [53]. The obtained Δ*E* values (Table 1) indicate that the WWS-IM and PWS-IM flours could easily be classified as having different colors (Δ*E* > 5), while the color differences between the WWS-IM and WS-RM flours could be perceived by an inexperienced observer, but without the ability to confirm whether they correspond to different colors (2 < Δ*E* < 3.5).

### 3.2. Batter Consistency

The use of flour obtained from polished and impact-milled grains (PWS-IM) led to batters that required more than three times the extrusion strength than to the other samples (Table 2). The relationship between batter consistency and the quality of the gluten-free bread reveals contradictions in the literature. Renzetti and Arendt [54] suggested that a decrease in batter consistency may improve batter development by reducing the resistance to expansion during proofing. Other authors stated that a more viscous batter consistency leads to bread with a higher specific volume (SV) because increasing the viscosity improves batter development and gas retention; in turn, these increase the bread volume [55,56].

### 3.3. Characterization of Bread

The specific bread volume ranged from 4.7 cm^3^/g to 4.8 cm^3^/g (Table 3); they showed no significant differences (*p* > 0.05). The values were higher than those reported by Angioloni and Collar [57] when using 40% white sorghum flour (3.2 cm^3^/g) as a replacement for wheat flour in breadmaking. Schober et al. [13] also reported significantly lower bread volumes (1.77 cm^3^/g–1.84 cm^3^/g) for loaves elaborated with white and red sorghum flours and corn starch in a 70/30 ratio. Probably, differences are attributed to the formulation, since those authors used corn flour, xanthan gum, and milk proteins, unlike the ones used in the present work.

The WS-RM sample revealed a marked depression in the central zone of the bread (Figure 2) compared to the other samples, probably due to the pressure exerted by gases and steam on the crumb during formation, causing the collapse of the structure [56,58]. Trappey et al. [31] found a decrease in the specific volume of gluten-free bread made with less refined sorghum flour (i.e., an increase in the fiber content).

In the present investigation, the bread made with WWS-IM, which had the highest ash content, showed neither a significant difference in the specific volume nor a greater central depression unlike WS-RM, which had a lower ash content. This can possibly be accounted for by analyzing the pasting profiles of the flours (Table 1). The final viscosity achieved by the roller-milled flour (WS-RM) was significantly higher than that of the other samples, indicating a higher proportion of starch available for yeast action, causing greater growth during fermentation. However, the structure did not hold up, resulting in its partial collapse during baking.

Crumb freshness is closely related to the specific structure of bread, particularly to the textural properties of the cell walls that form the air cells in the bread [59,60]. Gluten-free bread batter is usually prepared with a much higher proportion of water, between 80 and 110% by flour, compared to about 65% for wheat flour dough. Water, in addition to hydrating starch granules, also plays a major role in bread aging; starch retrogradation requires incorporating water into the crystalline structure, reducing the amount of it available to facilitate intermolecular interactions [61]. This has a significant effect on crumb firmness as the loss of these interactions decreases cohesion and leads to crumbling [62].

A textural analysis of the bread samples (Table 3) evidenced significant differences, especially in hardness, gumminess, and chewiness, The WS-RM bread crumb showed the greatest hardness and gumminess, while the PWS-IM bread crumb revealed the lowest values for these parameters and also for chewiness. Abdelghafor et al. [63] reported that bread made with a higher proportion of sorghum as a partial replacement for wheat flour presented an increase in hardness, which was higher in breads made with whole-grain sorghum flour, yet no changes in chewiness were observed.

An analysis of the distribution of alveoli can indicate whether there was an adequate incorporation of air during beating and whether the structure formed was able to retain it during baking. The predominant mechanism for forming air cells, or alveoli, in gluten-free bread occurs during beating. Carbon dioxide, produced as a by-product of yeast fermentation, diffuses into the air cells formed and causes them to expand [31]. Additives such as hydrocolloids and emulsifiers are included in the formulation to improve batter development and gas retention by increasing the viscosity of the system and interacting with different flour and batter components [21,56,64,65,66].

The WWS-IM and WS-RM bread samples showed greater percentages of the crumb area occupied by cells (air), which were smaller (*p* < 0.001) than in the bread made with the PWS-IM flour (Table 3). The WS-RM crumb had the lowest mean cell size, explaining why it had the highest crumb hardness. Additionally, as the extrusion resistance of the batters increased, the total number of cells decreased, as did the area they occupied.

The bread made with the PWS-IM flour had the largest cell size, which was linked to lower hardness, gumminess, and chewiness values (*p* < 0.001). In addition, this was the only bread that showed no collapse of its structure, which would indicate better gas retention during baking.

Figure 2 shows a uniform distribution of the alveoli in all the samples; this is reinforced by the statistical analysis, which shows no significant differences in this parameter (*p* > 0.05). Schober et al. [13] reported that sorghum breads with higher numbers of cells had greater hardness, while the softer breads presented larger alveoli.

High-quality bread is characterized by a high level of porosity and an open and homogeneous crumb structure [59]. Although several studies have evaluated the influence of raw materials on the quality of gluten-free bread [18,67], few have highlighted the effect of particle size. This parameter has proven relevant in making gluten-free bread [8,16]. The analysis of the average particle size (D4.3) evidenced significant differences between the flours. The PWS-IM flour had the highest particle size (D90 and D4.3), and WWS-IM had the highest polydispersity index (*p* = 0.0365), indicating greater heterogeneity in particle size, probably due to the presence of bran milled less homogeneously than the endosperm. Flour with a larger average particle size was associated with a smaller gas cell/total area ratio, yet those alveoli were larger, as observed in the sample formulated with the PWS-IM flour. It is important to note that only three flours were analyzed, so it is necessary to assay a higher number of flours to confirm these results.

The WWS-IM bread crumb had the highest *a** and the lowest *L** values, in agreement with a darker crumb in the brown range. The lightest color was acquired by the bread elaborated with PWS-IM, with significantly higher *L** and lower *a** values (*p* < 0.001) (Table 3). The presence of bran and germ, indicated by a higher ash content (Table 1), in WWS-IM gave a darker and browner color to both flours and their respective breads.

The PWS-IM bread crumb had a higher Δ*E* than the WS-IM bread crumb (Table 3) compared to WWS-IM. Both the PWS-IM and WWS-IM bread crumbs obtained different colors from the WWS-IM crumb, and this could be identified by an inexperienced observer (Δ*E* > 5) [53]. Yousif et al. [68] indicated that the substitution of up to 50% of wheat flour with whole grain white sorghum flour in the elaboration of bread led to an *L** value like those made entirely from wheat (*L*: 70–72). This implies greater acceptability by traditional consumers who value the whitish appearance of bread [69]. However, the authors reported that samples of bread made with the addition of sorghum flour in varying proportions (between 30% and 50%) were weighted significantly better in a sensory analysis test than those of the control (100% wheat flour). In this study, the use of 83% PWS-IM flour allowed us to obtain bread with an *L** value (63.9) similar to those described by Yousif et al. [68] with the addition of only 30% white sorghum flour.

However, it should be noted that in recent years, consumer preferences have been changing toward bread made with whole grain flour, which provides higher fiber and mineral contents despite being darker, as they are highly valued [70]. Fiber consumption, which is frequently inadequate in individuals adhering to a gluten-free diet, can alter the pace of nutrient digestion and/or assimilation, particularly for substances like starch and proteins [71]. Hence, greater emphasis is placed on the nutritional status of celiac patients, given that carbohydrates, proteins, and lipids are often consumed in unbalanced proportions, while the intake of some essential nutrients is often low [25,26].

### 3.4. In Vitro Digestibility of Bread

Initially, the polyphenol content and antioxidant activity of the flours and their respective breads were analyzed in methanol/water (M/W) extracts (Table 4). Moreover, the phenolic profile was determined via HPLC-MS/MS, and 15 compounds were tentatively identified in the samples (Appendix A): fourteen phenolic derivatives belonging to the families of polyamines, phenolic acids, and flavonoids and one hydroxy fatty acid. They were identified according to their exact mass, fragmentation pattern, and elution order by comparing them with information from previous publications [38,39,41,67]. All compounds were identified in flours and breads.

To provide a health benefit, the compounds of interest present in food must first be bioaccessible [72]. This depends on the digestion and release of the compounds from the food matrix, making them accessible for absorption. In the present work, in vitro digestion was performed to evaluate the potentially bioaccessible and dialyzable fractions of the polyphenols and their antioxidant activities. The determination of nutritionally relevant minerals during the digestion process was also performed. We used bread previously made with whole grain sorghum flour (WWF-IM) and polished flour (PWS-IM), both of which were prepared via impact milling.

The WWS-IM flour showed the highest polyphenol and antioxidant capacity values, as determined via the FRAP and ABTS (Table 4). As mentioned before, polyphenols are mostly found in the outer layer covering the grain [73]; thus, it was expected that the flour that fully retained the bran would have a greater presence of polyphenols. In general, the values agreed with those reported by Palavecino et al. [36] for the TPC in whole white sorghum flour and were higher than those reported by Rao et al. [46], who used the same type of grains. This variation could be ascribed to factors such as differences in sorghum varieties, extraction procedures, and/or extraction solvents used [74].

The bread M/W extracts displayed a pattern similar to that of the flours, with a higher TPC in the whole-grain sample and a higher antioxidant capacity. However, the TPC values in the bread were lower than those described by Yousif et al. [68] for bread made with 40% white sorghum flour (0.49 ± 0.02 mg GAE/g), which also agrees with the lower antioxidant capacity than the capacity reported by these authors.

The bread samples were subjected to an in vitro digestion assay to evaluate the TPC and antioxidant activity at different stages (Table 4). Although in vitro techniques show no biological effect, they can provide insights into the antioxidant effects of extracts and their chemical mechanisms of action [75]. The dialyzable fraction corresponds to the polyphenols that can be absorbed via passive diffusion. After the bread samples underwent chewing and gastric digestion, intestinal digestion was simulated, and samples were collected from the dialyzable fraction (M1), the non-dialyzable intestinal fraction (M2), and the dialyzable fraction after colonic fermentation (M3).

In general, during all the digestion stages, the polyphenol content exhibited a similar trend to that of the extracts, with significantly higher levels (*p* < 0.05) found in the breads made from whole flour at almost all stages. However, when it came to antioxidant and radical activity, the variances were minor.

Intestinal absorption (M1) determines potential bioavailability. The TPCs in all samples decreased significantly after intestinal digestion when compared to the pre-digestion M/W bread extracts. This could be attributed to the variation in TPC extractions seen using organic solvents in relation to those resulting from human digestion [76]. Additionally, pH changes could lead to the instability and hydrolysis of compounds released in the early stages by digestive enzymes.

The reducing capacity of bread also decreased significantly in all samples considered bioaccessible after intestinal digestion (M1). The reducing capacity values showed no significant differences among them. However, the antiradical activity increased significantly at this stage compared to the pre-digestion extracts. One possible explanation for this is that polyphenols might experience depolymerization during processing/digestion, leading to the possibility that certain compounds could improve or diminish their initial antioxidant activity or even lose it [77].

The TPC in the unabsorbed fraction of the intestinal digesta (M2) significantly decreased in the extracts; however, its values were greater than those observed in M1. The reducing capacity of M2 was lower than that in the undigested bread but higher than the capacity found in the intestinal absorption stage. Antiradical activity was not detected at this stage.

After colonic fermentation, the polyphenol content in the bioaccessible fraction (dialyzable, M3) was significantly lower than the content found in the extracts but higher than the content found in the intestinal absorption fraction (M1), suggesting a positive effect of the intestinal microbiota on the bioavailability of phenolic compounds. This may be attributed to pH variations and enzymatic activities during digestion which could affect cell wall integrity and release undetected antioxidant compounds and hydrolyzed proteins, resulting in peptides with antioxidant properties [78,79]. These compounds, which are excreted, can be fermented in the colon by bacteria, and a fraction may be absorbed [80].

The M3 bread digests showed a significantly lower reducing capacity than the reducing capacity found after intestinal digestion (non-dialyzable fraction, M2), yet values were particularly low in both cases. Regarding antiradical activity, after colonic fermentation (bioaccessible fraction, M3), these values were significantly lower than those seen in all previous stages except for the fraction not absorbed after intestinal digestion, in which it was not detected.

The disparity observed between the FRAP and ABTS values can be ascribed to the distinct mechanisms assessed by both assays. [81]. Similar trends were reported by Podio et al. [82] with respect to whole-wheat flour paste, observing a correlation between the TPC and reducing capacity through in vitro digestion but not with antiradical activity.

### 3.5. Evaluation of Mineral Content

Sorghum is known to be a good source of minerals, with potassium (K), phosphorus (P), and magnesium (Mg) being the most abundant, followed by nutritionally important microelements such as copper (Cu), iron (Fe), manganese (Mn), and zinc (Zn) [83,84]. However, the mineral profile of sorghum can vary depending on the variety, growing conditions, and location [85,86,87].

In this study, Cu, Fe, Mn, and Zn contents were measured in the flour and bread samples before and after in vitro digestion. The elements were also quantified in the fraction absorbed in the small intestine (dialyzable, M1) because most of the minerals are absorbed in the small intestine [88].

The contents of Cu, Fe, Mn, and Zn in the flour and bread extracts are shown in Table 5. In general, the mineral content was higher in the WWS-IM flour than in the PWS-IM flour. The Mn and Zn content values were similar to those reported in previous studies [89], while Cu showed lower values. Fe concentrations were lower than those reported by Gerrano et al. [90] but similar to those provided by de Morais Cardoso et al. [91] for whole sorghum flour.

Compared to their respective flours, the bread samples showed lower mineral contents, which could be attributed to the baking process and/or bread formulation. The proportion of flour in the recipe was 80% of the total ingredients. The WWS-IM bread had significantly higher concentrations of all analytes than the bread made with partially refined flour (PWS-IM).

Considering the recommended daily allowance (RDA) for an adult person, for each mineral (mg) analyzed (Cu: 0.9; Fe: 8; Mn: 2.3; and Zn: 11), we estimated the RDA (%) covered by a serving of sorghum bread (Figure 3).

As expected, the WWS-IM bread covered significantly higher percentages of the RDAs for all minerals than the bread made with partially refined flour (WWS-IM). On average, servings of bread made with WWS-IM and PWS-IM covered approximately 14.42% and 9.35% of the RDAs for Cu, 11.7% and 7.08% for Fe, 53.88% and 46.75% for Mn, and 10.44% and 4.75% for Zn, respectively (Figure 3). Note the contribution of Fe and Zn to the overall mineral intake since a serving of whole bread alone could cover approximately 10% of the daily requirement for these essential micronutrients. Deficiencies in Fe and Zn are major public health concerns worldwide, particularly in food-insecure countries [92]. Regarding Mn, both the WWS-IM and PWS-IM breads contributed nearly 50% of the RDA for Mn, with a slightly higher contribution made by the whole bread.

To determine the potentially absorbable minerals during intestinal digestion, we assessed the dialyzability (%) of the breads after subjecting them to in vitro digestion. The nutritional quality of sorghum is influenced by its chemical composition and the presence of anti-nutritional factors such as phytates. These compounds are the primary storage form of phosphate and are commonly found in plants, particularly in cereals and legumes. Phytates have been shown to interact with proteins, vitamins, and various minerals, limiting their bioavailability in humans and animals [93].

The minerals with the highest dialyzability were Cu and Fe. In turn, for both minerals, the dialyzability was higher in the bread extracts prepared from the PWS-IM flour than in those elaborated from the WWS-IM flour, although the differences were only significant for Cu (Table 6). The bioavailability of non-heme iron from vegetables and legumes typically ranges from 5 to 12%, which is lower than the values found in this study. The bioavailability of Fe depends on the reduction of ferric iron (Fe^3+^) into ferrous iron (Fe^2+^) in the intestinal lumen, which is facilitated by gastric acid and ferric reductases [94].

Regarding Mn, despite its significant contribution to the daily dietary reference intake (DDR) based on its content in the bread, it was not detected in the sample made with the WWS-IM flour. However, it was found in the sample made with the PWS-IM flour, but at a very low percentage, indicating a potentially poor absorption during intestinal digestion. Manganese is a trace element that is crucial for cellular protection against free radicals, as well as for the synthesis of glycosaminoglycans and cholesterol [95], making meeting the suggested daily requirements important.

The behavior of Zn was similar to that of Fe, and although an increase in crosstalk was observed in the PWS-IM sample, differences were not statistically significant. These findings agree with those of various studies that have shown similar trends in products made with white and whole-wheat flour [96,97] and bread made with whole quinoa flour [98]. This could be attributed to the particular composition of bread concerning the quantity and quality of proteins, along with the existence of substances like fibers, polyphenols, and phytates, which may inhibit the bioavailability of minerals. Moreover, the chemical forms of elements and interactions with nutrients may also contribute to this effect [99].

## 4. Conclusions

The study showed that the milling process used to obtain sorghum flour significantly/notably influenced the characteristics of the bread produced. The partially refined flour milled via impact revealed the highest batter extrusion force, indicating differences in its water absorption properties compared to the other flours. This flour produced bread with a soft crumb, fewer but larger alveoli in the crumb, and a structure that did not collapse.

An evaluation of the content of polyphenolic compounds and their antioxidant activity showed that the bread made with whole sorghum flour yielded the highest value for both determinations. Whole sorghum flour also showed the highest values for nutritionally relevant minerals such as Cu, Fe, Mn, and Zn. The same trend was observed in breads made with this flour, although with lower mineral contents when compared to their respective flours.

In vitro digestion tests showed that the total polyphenol content decreased significantly in all samples during digestion, with the whole flour bread maintaining significantly higher values than the breads obtained from partially refined flour. During the colonic fermentation stage, the total polyphenol content was higher than after intestinal digestion, possibly due to the greater dialyzability of the compounds or the positive effect of intestinal microbiota on bioaccessibility. In turn, the antiradical activity increased significantly during the intestinal digestion stage with respect to the bread extracts, indicating the release of phenolic compounds and greater antioxidant capacity.

The evaluation of potentially absorbed minerals revealed that a portion of bread made with whole sorghum flour would cover a higher percentage of the RDA of all the minerals analyzed, with Mn reaching 50%. However, the dialyzability of minerals was higher in the breads from partially refined flour, especially Fe and Zn, suggesting that a greater presence of bran could influence their absorption.

## Figures and Tables

**Figure 1 foods-12-03030-f001:**
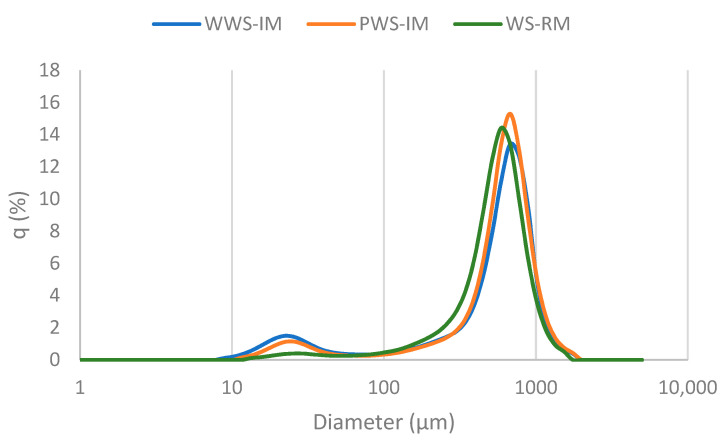
Particle size distribution patterns of sorghum flour. WWS: whole white sorghum flour; PWS: polished white sorghum flour; IM: impact milling; RM: roller milling.

**Figure 2 foods-12-03030-f002:**
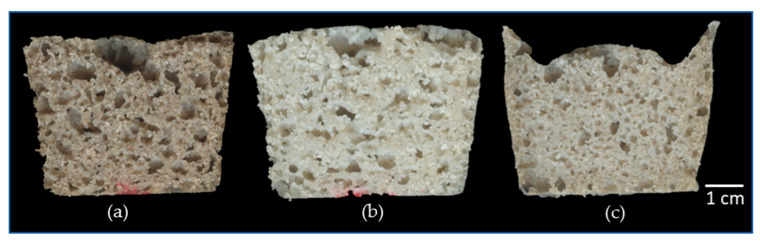
Photographs of sorghum bread samples produced with whole white sorghum flour (**a**) (WWS-IM), polishing plus impact milling flour (**b**) (PWS-IM), and roller milling plus sieving flour (**c**) (WS-RM).

**Figure 3 foods-12-03030-f003:**
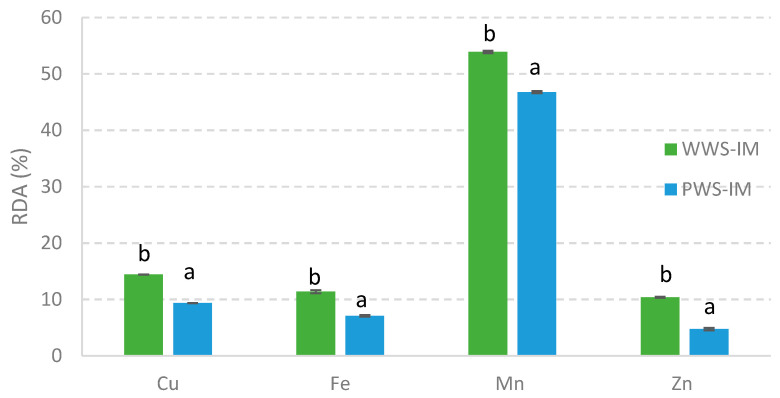
Percentage of the recommended daily allowance (RDA) of minerals covered by a serving portion of sorghum bread. Based on a portion of 120 g of bread for a diet of 2000 Kcal/day. Values are expressed as means (n = 3) ± their standard deviations. Different letters in the column for the same mineral indicate significant differences (*p* < 0.05). WWS-IM: bread made with whole grain white sorghum flour; PWS-IM: bread made with polished white sorghum flour. IM: impact milling.

**Table 1 foods-12-03030-t001:** Flour characteristics.

Sample	Yield (%)	Ash (%)	*L**	*a**	*b**	Δ*E*	VP (cP)	BD (cP)	VF (cP)	SB (cP)	TP (°C)
WWS-IM	99.2 ^c^	1.43 ^c^	76.06 ^a^	1.73 ^c^	15.65 ^a^	-	924 ^a^	63 ^a^	1966 ^a^	1104 ^a^	90.1 ^b^
PWS-IM	72.4 ^b^	0.54 ^a^	87.50 ^c^	0.16 ^a^	15.71 ^a^	11.54 ^b^	1115 ^b^	133 ^c^	2338 ^b^	1358 ^b^	90.0 ^b^
WS-RM	51.6 ^a^	0.71 ^b^	79.84 ^b^	1.22 ^b^	14.93 ^a^	3.89 ^a^	1851 ^c^	104 ^b^	2988 ^c^	1240 ^ab^	86.4 ^a^

Values are expressed as mean (n = 3). Different letters in the same column express significant differences (*p* < 0.05). WWS: whole white sorghum flour, WS: white sorghum flour; IM: impact milling; RM: roller milling; P: polishing; Δ*E*: color difference; VP: peak viscosity; BD: breakdown; VF: final viscosity; SB: setback; TP: pasting temperature; TP: pasting temperature.

**Table 2 foods-12-03030-t002:** Extrusion force for the different batters.

Sample	Extrusion Force (N)
WWS-IM	0.858 ^a^
PWS-IM	3.306 ^b^
WS-RM	0.731 ^a^

Values are expressed as means (n = 2). Different letters in the same column express significant differences (*p* < 0.05). WWS: whole white sorghum flour; WS: white sorghum flour; IM: impact milling; RM: roller milling; P: polishing.

**Table 3 foods-12-03030-t003:** Technological characteristics of bread.

Sample	WWS-IM	PWS-IM	WS-RM
VE (cm^3^/g)	4.74 ± 0.3357 ^a^	4.68 ± 0.31 ^a^	4.78 ± 0.03 ^a^
Hardness (N)	20.31 ± 1.7319 ^b^	14.87 ± 1.39 ^a^	25.87 ± 3.26 ^c^
Gumminess (N)	15.33 ± 3.25 ^b^	10.96 ± 2.59 ^a^	25.74 ± 3.62 ^c^
Cohesiveness (J)	1.08 × 10^−4^ ± 4.75 × 10^−5 a^	8.95 × 10^−5^ ± 2.09 × 10^−5 a^	8.88 × 10^−5^ ± 1.53 × 10^−5 a^
Chewiness (J)	1.91 × 10^−3^ ± 1.04 × 10^−3 b^	1.33 × 10^−3^ ± 3.41 × 10^−4 a^	2.31 × 10^−3^ ± 5.79 × 10^−4 b^
*L**	45.77 ± 4.47 ^a^	63.94 ± 3.04 ^c^	54.01 ± 1.92 ^b^
*a**	4.78 ± 0.34 ^c^	1.10 ± 0.28 ^a^	2.84 ± 0.86 ^b^
*b**	18.70 ± 1.09 ^b^	18.30 ± 0.28 ^ab^	17.39 ± 1.28 ^a^
Δ*E*	-	20.11 ± 4.92 ^b^	6.88 ± 2.83 ^a^
Number of cells/mm^2^	206.50 ± 9.50 ^b^	113.50 ± 2.50 ^a^	253.50 ± 2.50 ^c^
Mean cell area (mm^2^)	1.37 ± 0.05 ^b^	2.11 ± 0.02 ^c^	1.03 ± 0.01 ^a^
Cell to total area ratio (%)	30.85 ± 0.32 ^b^	26.49 ± 0.95 ^a^	28.92 ± 0.54 ^ab^
Uniformity	4.88 ± 0.09 ^a^	3.76 ± 0.76 ^a^	5.86 ± 0.74 ^a^

Values are expressed as means (n = 3). Different letters in the same row correspond to significantly different values (*p* < 0.05). VE: specific volume; WWS: whole white sorghum flour; PWS: polished white sorghum flour; IM: impact milling; RM: roller milling; Δ*E*: color difference.

**Table 4 foods-12-03030-t004:** Determination of polyphenol content and antioxidant activity before and after in vitro digestion process.

Stage	Sample	TPC (mg GAE/100 g)	FRAP (mmol Trolox/100 g)	ABTS (mg TE/100 g)
Flour extract	WWS-IM	53.63 ± 2.54 ^e B^	0.29 ± 0.03 ^e A^	13.67 ± 0.49 ^c A^
	PWS-IM	21.29 ± 0.06 ^c A^	0.18 ± 0.02 ^cd A^	7.89 ± 0.61 ^b A^
Bread extract	WWS(B)-IM	41.20 ± 2.05 ^d B^	0.26 ± 0.02 ^e A^	13.52 ± 1.77 ^c A^
	PWS(B)-IM	18.50 ± 0.63 ^c A^	0.13 ± 0.02 ^bc A^	7.83 ± 0.18 ^b A^
Dialyzable fraction (M1)	WWS(B)-IM	0.43 ± 0.04 ^a B^	0.02 ± 0.00 ^a B^	18.35 ± 0.38 ^d B^
	PWS(B)-IM	0.25 ± 0.09 ^a A^	0.01 ± 0.00 ^a A^	17.54 ± 0.48 ^d A^
Non-dialyzable fraction (M2)	WWS(B)-IM	5.63 ± 0.66 ^ab A^	0.19 ± 0.02 ^d B^	ND
	PWS(B)-IM	7.21 ± 0.80 ^b B^	0.14 ± 0.02 ^bc A^	ND
Colonic fermentation (M3)	WWS(B)-IM	2.46 ± 0.08 ^ab B^	0.11 ± 0.01 ^b B^	1.23 ± 0.19 ^a A^
	PWS(B)-IM	0.84 ± 0.09 ^b A^	0.02 ± 0.00 ^a A^	1.76 ± 0.36 ^a A^

Values are expressed as means (n = 4) ± standard deviations. Different lowercase letters in the same column indicate significant differences (*p* < 0.05) between samples throughout the process. Different capital letters in the same column indicate significant differences between samples (*p* < 0.05) at the same stage. TPC: total polyphenols; WWS-IM: whole white sorghum flour; PWS-IM: polished white sorghum flour. IM: impact milling; RM: roller milling. ABTS: 2,2′-azino-bis (3-ethylbenzothiazoline-6-sulfonic acid); FRAP: ferric reducing antioxidant power.

**Table 5 foods-12-03030-t005:** Mineral content in sorghum flour and bread.

Sample		Cu (µg, g^−1^)	Fe (µg, g^−1^)	Mn (µg, g^−1^)	Zn (µg, g^−1^)
Flour	WWS-IM	1.45 ± 0.02 ^a^	12.44 ± 0.26 ^b^	16.80 ± 0.95 ^a^	11.63 ± 0.55 ^b^
	PWS-IM	0.97 ± 0.13 ^a^	7.43 ± 0.01 ^a^	14.29 ± 1.36 ^a^	4.85 ± 0.79 ^a^
Bread	WWS-IM	1.08 ± 0.03 ^b^	7.58 ± 0.28 ^b^	10.33 ± 0.20 ^b^	9.53 ± 0.10 ^b^
	PWS-IM	0.70 ± 0.04 ^a^	4.72 ± 0.15 ^a^	8.96 ± 0.20 ^a^	4.36 ± 0.22 ^a^

Values are expressed as means (n = 3) ± their standard deviations. Different lowercase letters in the same column indicate significant differences (*p* < 0.05) between flour and bread, respectively. WWS-IM: whole white sorghum; PWS-IM: polished white sorghum; IM: impact milling.

**Table 6 foods-12-03030-t006:** Dialyzability (%) of minerals in bread during the in vitro digestion process.

Sample	Cu (%)	Fe (%)	Mn (g)	Zn (g)
WWS-IM	22.25 ± 1.63 ^a^	26.50 ± 0.65 ^a^	ND	1.85 ± 0.44 ^a^
	(0.24)	(7.58)		(0.18)
PWS-IM	29.95 ± 2.03 ^b^	29.53 ± 5.81 ^a^	1.12 ± 0.31	2.33 ± 0.09 ^a^
	(0.22)	(4.72)	(0.10)	(0.10)

Values are expressed as means (n = 3) ± standard deviations. Different letters in the same column indicate significant differences (*p* < 0.05). WWS: whole white sorghum flour bread; PWS: polished white sorghum flour bread; IM: impact milling. Values in parentheses indicate concentration in µg/g of dialyzed minerals. Different letters in the same column indicate significant differences (*p* < 0.05).

## Data Availability

The data presented in this study are available on request from the corresponding author.

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
