# Peer review of "Sorghum (Sorghum bicolor L. Moench) Gluten-Free Bread: The Effect of Milling Conditions on the Technological Properties and In Vitro Bioaccessibility of Polyphenols and Minerals"

_foods, 2023, doi:10.3390/foods12163030_

Round 1

Reviewer 1 Report

Comments and Suggestions for Authors

Authors have performed a study on GF sorghum breads and undertaken a detailed characterization of three breads from three different flours. But a limitation is that they make some statements for which three flours are not sufficient. E.g. there are 2 flours, but they do not only differ by particle size, but also by chemical composition. See also later the comment at results. In the whole MS this fact has not been addressed sufficiently. Thus, some of the results are interesting, e.g. the nutritional (mineral and digestibility) results but overall, they are of limited value to deliver explicit information for future sorghum gf baking. I recommend to sharpen the discussion in this respect.

Abstract:

Line 23: D 4,3 – please mention full words, this abbreviation is not common and clear.

Line 26, 27 – “

The in vitro 26 digestibility of bread decreased the content of total polyphenols and their antioxidant activity during the digestion steps”??? should it be like this: The in vitro 26 digestibility of bread WAS decreased BY the content of total polyphenols and their antioxidant activity during the digestion steps?

Line 28 – in “a portion of” bread – please specify the type of breads… or the amount of sorghum…

Line 30: replace greater by higher

Introduction:

Very informative and concise. High number of relevant citations (could have been reduced, but you can well leave it here)

Line 35: name all abbreviations in full at first mention – her TACC =?

Objective is generally good, but could be described somewhat more explicitly.

Materials and methods:

All materials (chemicals etc.) need to be specified (name, type, source, supplied) within the methods.

Line 126: unpolished sample, means wholegrain containing the bran or even containing the husk as well?

Line 133: particle size distribution needs more detailed explanation: can you describe what is D (4,3)? Also unclear is the description “diameters were 10% (D10), 50% (D50), and 90% (D90) lying below these (which?) values?

Line 141: Perten is now Perkin Elmer USA

Line 185: how did you determine elasticity and cohesiveness?

Line 196: total cell area is giving no information, in particular when you have not mentioned the size of area (please add this information) that was measured. Total cell area is need to calculate then the ratio (cell fraction)

Line 197: Grain uniformity was determined as the ratio of small to large cell counts (i.e., the ratio of the number of cells with areas lower than 4 mm2); this is unclear: ratio of number of cells below 4mm² to number of cells that were larger? Why is this ratio corresponding to uniformity? I think the best figure to take for uniformity is the standard deviation of the mean cell area: the higher this is the lower is the uniformity.

Line 202: radical scavenging activity was not measured, at least I see not results?

Results and Discussion

Chemical analyses (at least protein and ash content) of flour would have helped to explain difference better.

Table 3: values should be given as mean plus/minus standard deviation

Line 343: with such differences in the flours, it is strange that volume was not influenced. In particular as all other values are different

Line 411-417: this influence of particle size might have been true for these three breads, but it is a not very reliable to make such a statement with only three types of flour and 3 breads, in particular as these flours are also very different in chemical composition. This has to be considered and discussed here.

Reviewer 2 Report

Comments and Suggestions for Authors

The research is quite interesting due to the fact that the gluten-free food sector can be one of the most profitable branches of the food industry, not only because of people suffering from celiac disease, but also because of the prevailing trend where a significant number of people switch to a gluten-free diet, which is an expression of the search for alternative ways of eating compared to the traditional one, which also justifies the purposefulness of the research undertaken. Nutritional and health-promoting, so that these products are preferred by consumers and attractive to them.

 The reviewed paper concerns Sorghum (Sorghum bicolor L. Moench) gluten-free bread: effect

of milling conditions on technological and nutritional properties

My first remark concerns the title of the work, in which the Authors talk about nutritional properties.

 In the methodology, they mean only ash and macronutrients. The nutritional composition includes protein , fat, carbohydrates, starch and sugars. Authors will focus on health-promoting compounds, i.e. on total polyphenols  and the profile of polyphenols as well as anti-oxidant properties (ABTS and FRAP) i.e pro-health properies

 In my opinion, sufficient INTRODUCTION, but please give more datails concerning polyphenols.

 In the methodology, please explain why the extracts were frozen to -80C?. Please provide the methodology for FRAP because it has not been described anywhere.

Parameters Lab are  currently the most popular way to describe color and is the basis of modern color management systems. The difference between two colors in the CIELab space is given by the equation for delta E.and is the usual Euclidean distance between two points in three-dimensional space.

It can be assumed that the standard observer notices the color difference as follows:

0 < ΔE < 1 - does not notice the difference,

1 < ΔE < 2 - only an experienced observer notices the difference,

2 < ΔE < 3.5 - the difference is also noticed by an inexperienced observer,

3.5 < ΔE < 5 - notices a clear color difference,

5 < ΔE - the observer has the impression of two different colors.

 Please calculate delta E and please add this into Section methods and Results ( concerning  flour and breads).

 In my opinion, in order to comprehensively examine the impact of grinding on the quality of obtained flours and breads, the Authors must determine protein, fat, carbohydrates, starch, and sugars.

 I am asking the Authors to recalculate FRAP because there are significantly low data here.

In results, please compare the parameter (if the percentage differs) in a given flour or bread, and not repeat the results that are already included in the tables or figures .

Conclusions too general, please enter meaningful data for them.

Comments on the Quality of English Language

The research is quite interesting due to the fact that the gluten-free food sector can be one of the most profitable branches of the food industry, not only because of people suffering from celiac disease, but also because of the prevailing trend where a significant number of people switch to a gluten-free diet, which is an expression of the search for alternative ways of eating compared to the traditional one, which also justifies the purposefulness of the research undertaken. Nutritional and health-promoting, so that these products are preferred by consumers and attractive to them.

Reviewer 3 Report

Comments and Suggestions for Authors

This research aims to study the effects of mill type (impact and roller) on flour properties and gluten-free bread quality. Bread was made with three white sorghum flours: integral and polished (partially dehulled), both obtained by impact milling, and flour obtained by roller milling with subsequent sifting. Flour fractions were characterized; loaves of bread were subjected to an in vitro digestibility test to evaluate their antioxidant properties and the potential bioavailability of minerals.

 The Introduction section provides background about the topic.

2. Materials and methods section

Why applied the back extrusion test to characterize the dough.

Since extrusion requires that the food flow under pressure, it seems reasonable to use it on food that will flow fairly readily under an applied force and not to use it on those foods that do not flow easily, such as bread, cake, cookies, breakfast cereals, and candy.

Line 291: InfoStat software (Version 13p) – please add the producer name, city, country

 The results of this investigation showed that the milling process used to obtain sorghum flour notably influenced the bread characteristics, providing insights into its nutritional profile, and also, on in vitro digestibility of bread.

The work contributes to the knowledge of the effects of mill type on flour properties and gluten-free bread quality by assessing their technological quality, antioxidant activity, and mineral content.

The Conclusion section was supported by the results. The references are properly selected.

 Minor comments:

Line 35:  please write the full name for TACC and between brackets the abbreviation TACC

Comments on the Quality of English Language

Kindly review the editing and grammar.

Author Response

Paper title: Sorghum (Sorghum bicolor L. Moench) gluten-free bread: effect of milling conditions on technological and nutritional properties and in vitro bioaccessibility of polyphenols and minerals”

 We greatly appreciate the thorough and thoughtful comments provided on our submitted article.

All the additions and changes were written in red (Track Changes) in the attached new version of the manuscript.

Editor:

We revised the manuscript to reduce the total similarity index and make it less than 30%.

Reviewer 3

Comments and Suggestions for Authors

Introduction

Line 35:  please write the full name for TACC and between brackets the

abbreviation TACC

Gluten-free was used instead of TACC.

Materials and methods:

Why applied the back extrusion test to characterize the dough. Since extrusion requires that the food flow under pressure, it seems reasonable to use it on food that will flow fairly readily under an applied force and not to use it on those foods that do not flow easily,

such as bread, cake, cookies, breakfast cereals, and candy.

The production of gluten-free breads involves obtaining a batter (fluid) instead of a dough (as in breads with wheat flour, with gluten), so, according to the reviewer, this technique allows a very good characterization of this type of systems, as verified in other publications (https://doi.org/10.1016/j.lwt.2021.112819; https://doi.org/10.1016/j.foodhyd.2019.105315; https://doi.org/10.1016/j.foodres.2018.05.070; https://doi.org10.1177/1082013219894109)

Line 291: InfoStat software (Version 13p) – please add the producer name, city, country

The information of InfoStat software was added.
